# SkipSR: Faster Super Resolution with Token Skipping

## Abstract

Diffusion-based super-resolution (SR) is a key component in video generation and video restoration, but is slow and expensive, limiting scalability to higher resolutions and longer videos. Our key insight is that many regions in video are inherently low-detail and gain little from refinement, yet current methods process all pixels uniformly. To take advantage of this, we propose SkipSR, a simple framework for accelerating video SR by identifying low-detail regions directly from low-resolution input, then skipping computation on them entirely, only super-resolving the areas that require refinement. This simple yet effective strategy preserves perceptual quality in both standard and one-step diffusion SR models while significantly reducing computation. In standard SR benchmarks, our method achieves up to 60% faster end-to-end latency than prior models on 720p videos with no perceptible loss in quality. Video demos are available at our project page.

## 1 Introduction

Diffusion transformers are the dominant paradigm in image and video generation, but due to the quadratic cost of self-attention, computational time grows steeply with resolution and sequence length. Furthermore, diffusion models typically require tens of steps to produce high-quality outputs. To handle this, a common design choice is *cascaded* diffusion (Ho et al., 2022; Saharia et al., 2022a; Gao et al., 2025), which first generates low-resolution images or videos, then uses a conditional diffusion model to upscale and refine them into high-resolution results with fewer steps. However, because of the larger input size at higher resolutions, diffusion-based super-resolution (SR) steps can be very slow, often dominating the computation time. Speeding these up is essential for high-resolution video generation and restoration.

Most prior work on addressing this issue focuses on reducing the number of diffusion steps. While successful, this leaves gains on the table since even a single diffusion step is still expensive on high-resolution video. Another recent line of work experiment with alternative attention mechanisms, such as windowed, sliding tile, or spatially sparse attention. These works typically restrict the attention mechanism to focus on nearby tokens, resulting in significant speedups.

A common assumption behind all these works is that they spend equal computation on every input patch. However, not every patch is created equal. Our key insight is that many videos contain *visually simple regions*: for example, a blue sky or a blurry background. This phenomenon is particularly pronounced in super-resolution tasks, where the low-resolution conditional input often includes extensive visually simple areas. These regions can be simply resized and placed directly in the output, since they do not need refinement like more detailed regions of the input do.

We present Skip Super Resolution (SkipSR), a method that uses this idea to accelerate video super-resolution. SkipSR uses a lightweight mask predictor to route only patches that require refinement through the transformer, while the remaining patches skip the model entirely. The transformer model maintain positional awareness with mask-aware rotary positional encodings, and the two patch groups are composed together at the output stage, resulting in perceptually indistinguishable outputs at a significantly lower computational cost.

We primarily evaluate on video SR tasks, given their considerable inference cost, and validate SkipSR on top of a state-of-the-art video SR model (Wang et al., 2025b;a). Our experimental results demonstrate its versatility in both multi-step and one-step diffusion paradigms. SkipSR significantly

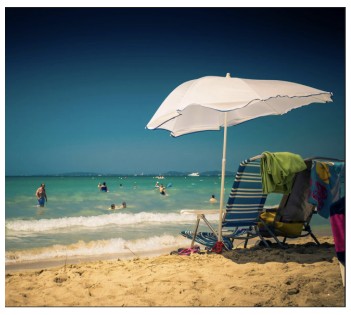 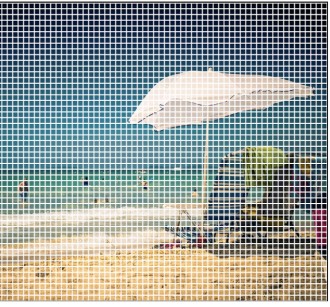 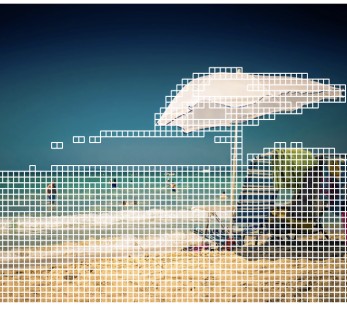

Original Image      Standard Diffusion Transformer      SkipSR (Ours)

Figure 1: **Patch Skipping.** Standard diffusion SR models refine the entire input, while SkipSR identifies and upscales only the patches that need refinement. This significantly reduces computation with no loss in perceptual quality.

accelerates super-resolution, yielding outputs that are perceptually indistinguishable from dense attention equivalents. In particular, we achieve up to $60\%$ faster super-resolution on 720p videos with no loss in quality, supported both by video-quality metrics and user studies, and reduce the diffusion time for 1080p videos by $70\%$.

In summary, the main contributions of our work are as follows:

- We demonstrate simple regions are common in videos, and that identifying and skipping these simple regions can match perceptual quality with a fraction of the compute.
- We propose SkipSR, consisting of a lightweight mechanism to identify complex regions and apply sparse attention only to them, leading to accurate and efficient super-resolution.
- We validate our hypotheses and design choices with extensive experiments, demonstrating a consistent speed-up while maintaining quality.

## 2 RELATED WORK

**Efficient Video Diffusion.** Video generation is prohibitively slow, with full-attention models like HunyuanVideo (Kong et al., 2024) taking tens of minutes to generate a 5-second video (Xi et al., 2025). Since most models require many steps (Song & Ermon, 2019; Ho et al., 2020; Meng et al., 2021), most work has focused on reducing the number of steps, through improving flow (Liu et al., 2022a; Geng et al., 2025), consistency losses (Song et al., 2023), or distillation (Salimans & Ho, 2022; Yin et al., 2024; Lin et al., 2025). We build upon these works by demonstrating speed-ups on both multi-step and single-step models.

Several methods have proposed speed-ups by increasing sparsity into the attention mechanism, thus reducing the number of tokens considered. Sliding window attention (Hassani et al., 2023; Liu et al., 2024; Zhang et al., 2025b) and other variants such as Radial Attention Li et al. (2025) restrict attention to focus on spatiotemporally local tokens. Other works builds on this by enforcing sparsity on attention heads in an online manner, such as SVG and Sparge (Xi et al., 2025; Zhang et al., 2025a). Our method differs from these by skipping the entire Transformer, rather than just the attention mechanism.

**Diffusion Models for Video Super-Resolution.** Alternatively, diffusion can be made more efficient by first diffusing at low resolution, then upscaling at the end. (Ho et al., 2022). Cascaded models like this are commonplace, such as SR3 (Saharia et al., 2022b), Stable Diffusion (Rombach et al., 2022), and Seedance (Gao et al., 2025). Diffusion is also considered standard for stand-alone video restoration. Recent works adapt image diffusers or condition video diffusers on LR inputs via ControlNet (Xu et al., 2023; Zhou et al., 2024; Wang et al., 2023b). While improving fidelity, these methods inherit the high cost of multi-step sampling and heavy conditioning. Parallel efforts aim to accelerate diffusion through fewer or single sampling steps, using rectified flows (Liu et al., 2022b), distillation (Zhang et al., 2023; Wang et al., 2025a; Huang et al., 2023; Chen et al., 2025),

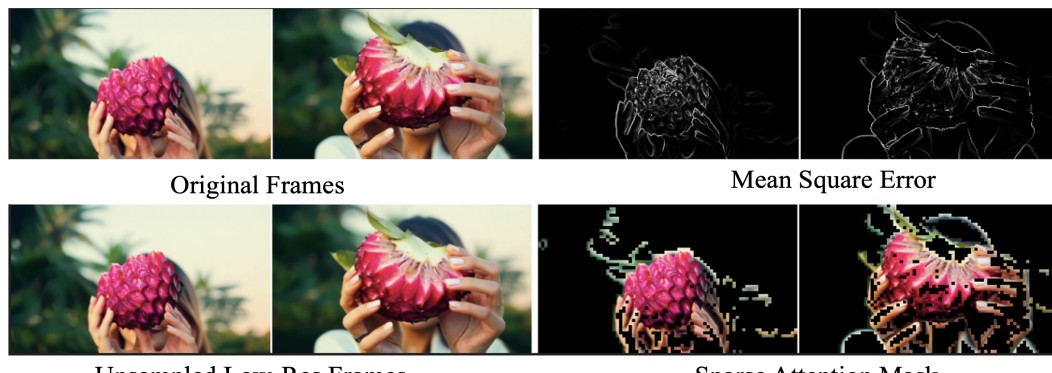

Original Frames · Mean Square Error

Upsampled Low-Res Frames · Sparse Attention Mask

Figure 2: **Oracle Mask Computation.** We identify the low-resolution video regions by comparing the original high-resolution input to a spatial downsampled, then upsampled version. This results in a attention mask, shown in dark, that excludes these regions from refinement.

or deterministic one-step models (Li et al., 2023). However, these methods still process every region of the input equally, so that simple videos take the same time as more complex ones.

**Efficient Super-Resolution.** Prior works have also attempted to accelerate super-resolution, mainly on images, by identifying which regions need less refinement. Sparse Mask SR (Wang et al., 2021) enforces sparsity to this end on CNN filters, while other works (Kong et al., 2021; Hu et al., 2022; Fan et al., 2024) apply different sub-networks to sub-regions of an input image based on their predicted complexity. Although these methods reduce FLOPS, they result in significantly slower wall clock time, as stated in the papers. YODA (Moser et al., 2025) uses attention masks to guide and improve diffusion for image SR, but does not accelerate the model. We provide a simple solution to these issues while significantly speeding up wall-clock inference time.

## 3 MOTIVATION AND ANALYSIS

**Selective Patch Processing.** High-resolution videos often contain regions that do not contain high-frequency details, e.g., a blue sky, a plain wall, or an out-of-focus background. We hypothesize that these regions can skip the expensive transformer computation and be upsampled by cheaper methods, such as bilinear interpolation, instead.

Concretely, we divide a high-resolution video $I$ of size $T \times H \times W \times 3$ into non-overlapping patches $P$ of size $4 \times 16 \times 16 \times 3$. We define a patch as *skippable* if, after area downsampling $D$ and bilinear upsampling $U$, the reconstruction mean squared error (MSE) is smaller than a threshold $\tau$:

$$\mathrm{MSE}\left(P, U(D(P))\right) \leq \tau \qquad (1)$$

Note that while there are many ways to classify the skippable regions, this is not the focus of our work. Empirically, we find this straightforward measurement suffices.

To measure the commonality of such skippable regions, we measure the percentage of patches satisfying the criterion in Eq. (1) with $\tau = 0.0002$ and a spatial downsampling factor of $4\times$ on a diverse set of videos. Furthermore, to validate skipping the patches does not degrade quality in the latent diffusion setting, we encode the original video $I$ through a VAE and swap the latent at the corresponding skippable region with the encoding of $U(D(I))$ before decoding back to the pixel space. Then, PSNR, SSIM, and LPIPS (Zhang et al., 2018) metrics are used to measure degradation against the original video $I$.

Table 1 shows the results of our analysis on videos from VBench (Huang et al., 2024), AIGC-30 (Wang et al., 2025b), VideoLQ (Chan et al., 2022), and YouHQ-40 (Zhou et al., 2024). In particular, VBench and AIGC-30 are AI generated videos, which yield the highest amount of skippable patches, up to 45%. For real-world videos, VideoLQ is a common SR evaluation dataset offered in low resolution. We use the upsampled version by SeedVR (Wang et al., 2025b) and yield 30% skippable patches. YouHQ has large camera movements and yields a lower 15% skippable patches.

Table 1: **Oracle Mask Analysis.** Using the oracle mask, we measure the reconstruction error of upsampling and swapping simple patches, as well as the resulting expected speedup. We spatially downsample by $4\times$ in all cases.

| Dataset | Resolution | Skippable % | PSNR↑ | SSIM↑ | LPIPS↓ | Speedup↑ |
|---|---|---|---|---|---|---|
| VBench | 720p | 44.8 % | 42.24 | 0.988 | 0.0400 | 1.8× |
| AIGC-30 | 720p | 41.9% | 43.11 | 0.989 | 0.038 | 1.7× |
| VideoLQ | 720p | 31.2% | 46.61 | 0.994 | 0.0239 | 1.5× |
| YouHQ-40 | 720p | 14.7% | 46.64 | 0.994 | 0.0149 | 1.2× |
| YouHQ-40 (corrupted) | 720p | 0.9% | 66.33 | 0.999 | 0.0002 | 1.0× |
| AIGC-30 | 1080p | 44.9% | 44.39 | 0.991 | 0.031 | 1.8× |
| VideoLQ | 1080p | 36.7% | 48.32 | 0.993 | 0.013 | 1.6× |

Overall, our approach can expect an average reduction of 30% tokens for regular super-resolution tasks while retaining a VAE reconstruction PSNR of 40+ and LPIPS of near 0, which are considered imperceptible to human eyes. However, as a limitation, we also show that the synthetically corrupted YouHQ contains heavy white noise everywhere. Such cases do not yield much patch savings and are less applicable to our approach.

**Estimating the Theoretical Speedup.** Next, we estimate the expected speed-up by completely skipping the patches from the transformer and profiling the diffusion model. We use the state-of-the-art SR method SeedVR2 (Wang et al., 2025a) as the baseline architecture, and the relative speed-up is provided in the last column of Table 1. This preliminary experiment shows that skipping patches can achieve $1.2\times$ to $1.8\times$ speed-up on the diffusion model. Inspired by these observations, our method is designed to (1) identify the simple regions from low-resolution input as accurately as possible and (2) produce high-quality super-resolution outputs from applying attention only on the complex regions.

## 4 METHOD

This section first provides a brief review of the video super-resolution model on which our method is built. We then introduce our approach in Section 4.2.

### 4.1 PRELIMINARIES

Recent diffusion-based SR methods adopt the latent diffusion paradigm (Rombach et al., 2022), where a pre-trained variational autoencoder (VAE) (Kingma & Welling, 2013) compresses images or videos into a latent space, and a diffusion transformer generates the high-resolution latent conditioned on text and low-resolution latents. During training, low-resolution inputs are obtained by applying synthetic degradations (e.g. blurring, downsampling, noise injection, compression) to high-resolution data. This procedure works well for super-resolution as well as more complex restoration. For more details, we refer the reader to Wang et al. (2025b).

We adopt the architecture used in SeedVR (Wang et al., 2025b), which combines shifted windows (Liu et al., 2021) and the native-resolution trick of Dehghani et al. (2023) and use MMDiT (Esser et al., 2024) for text-conditioning in the DiT. We employ a causal video VAE that compresses the pixel-space input by $8\times$ spatially and $4\times$ temporally, and the DiT operates on $1 \times 2 \times 2$ patches in latent space.

### 4.2 SPARSE SUPER-RESOLUTION

**Skip Prediction Model.** Unlike our preliminary analysis in Section 3, at inference time, we do not have a high-resolution ground-truth video to identify skippable patches. We train a lightweight predictor network to predict skippable patches given low-resolution videos. Specifically, our predictor network operates in the VAE latent space and is composed of 4 layers of 3D convolution and ReLU activations. The first convolution has a spatial stride of 2 to match teh patch size. The net-

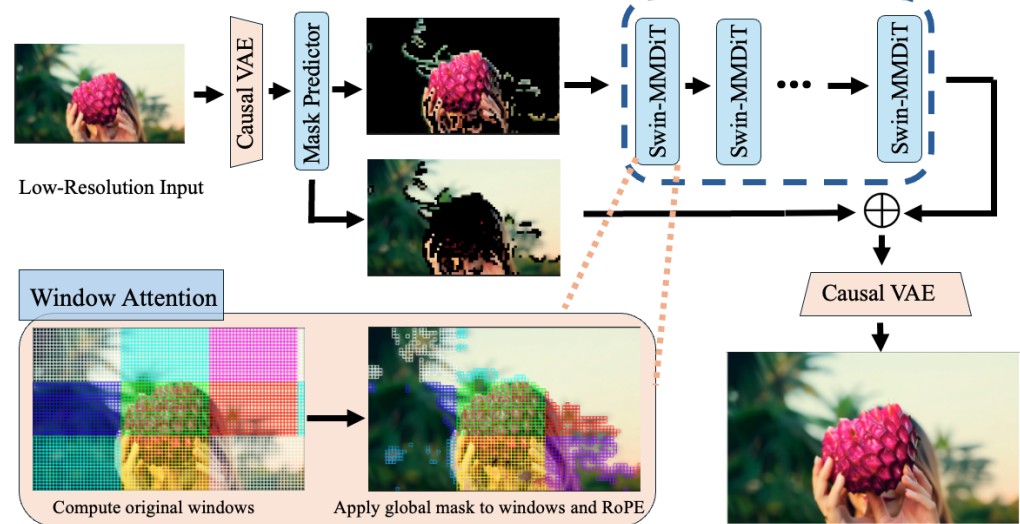

Figure 3: **SkipSR Overview.** We take as input low-resolution videos, project into latent space, then compute the complexity mask. Simple patches skip computation and are routed around the transformer, then composed with the refined output.

work takes latent of shape $t \times h \times w \times 16$ as input, and outputs binary classification logits of shape $t \times h/2 \times w/2 \times 1$. This corresponds to a patch size of $1 \times 2 \times 2$ in the latent space and $4 \times 16 \times 16$ in the pixel space.

During training, given high-resolution videos $I$ and their synthetic low-resolution pairs $D(I)$, we encode $D(I)$ through VAE and provide it as input to the predictor. The predictor network is trained using binary cross-entropy loss, where the ground-truth classification target is given by Eq. (1). During inference, low-resolution videos $i$ are used instead of $D(I)$, and the predictor network classifies skippable patches with high accuracy.

**Skip-Aware Diffusion Model.** Prior works that accelerate diffusion models modify the attention algorithm, but leave the other components, such as the FFN and LayerNorm, unchanged. Our key finding is that the predicted skipped tokens *can skip the Transformer altogether*. Typically, these skipped tokens are in the background, while the un-skipped tokens come from the more detailed foreground. We speculate that the un-skipped tokens, along with the text conditioning, provide enough context to produce high-quality super-resolutions.

After the mask predictor produces a mask $M$, we apply $M$ to the set of input latent patches, $P$. $P$ has $N$ patches, where $N = t \times h/2 \times w/2$, indexed as $P = \{p_1, p_2, \ldots p_N\}$ We then apply $M$ to $P$, partitioning it into two sets $P_{\text{skip}}$ and $P_{\text{unskip}}$. $P_{\text{skip}}$ receives no further computation.

$P_{\text{unskip}}$ is next passed through the Transformer's patch projection layer, and proceeds through each Transformer block. However, as shown in Figure 3, the patches in $P_{\text{unskip}}$ are no longer spatiotem-porally contiguous. For example, $P_{\text{unskip}}$ could consist of $\{p_3, p_4, p_1 3 \ldots p_N\}$. We thus need to ensure that the Transformer is aware of their relative positions. We accomplish this by modifying the rotary positional encodings (RoPE) (Su et al., 2021) in the attention operation. In RoPE, the embeddings in each token are rotated by an angle $\theta$ that is scaled by the position index:

$$\text{RoPE}(p_i, i) = \mathbf{R}_i \cdot p_i \tag{2}$$

where $p_i$ is the $i$-th patch in $P$, $d$ is the feature dimension, and $\mathbf{R}_m \in \mathbb{R}^{d \times d}$ is a block-diagonal rotation matrix with angles $i \cdot \theta_k$ for each dimension pair $k$, where $\theta_k = 10000^{-2k/d}$.

Rather than use the index $m$ of each patch in $P_{\text{unskip}}$, we use each the original index from $P$, encoding the relative position of each patch. With our previous example, if the first element of $P_{\text{unskip}}$ was $p_3$, we would apply RoPE with $i = 3$ rather than $i = 1$, ensuring that patches in $P_{\text{unskip}}$ are aware of their relative distances. After running the Transformer and the patch output projection

Table 2: **Main Quantitative Results.** We compare SkipSR's super-resolution output against SeedVR and SeedVR2 baselines on real-world (VideoLQ) and AI-generated (AIGC30) benchmarks. The best value is in **blue**, and the second-best in red. SkipSR matches the performance of SeedVR and SeedVR2 with significantly less compute.

| Dataset | Method | NIQE↓ | MUSIQ↑ | CLIP-IQA↑ | DOVER↑ | BRISQUE↓ | TFLOPs↓ | s/step↓ | Skipped% |
|---------|--------|-------|--------|-----------|--------|----------|---------|---------|----------|
| VideoLQ | SeedVR-3B | **4.069** | **57.41** | **0.318** | **8.009** | 35.45 | 2001 | 6.41 | 0 |
| | SeedVR-7B | 4.933 | 48.35 | 0.258 | 7.416 | **28.120** | 3139 | 7.74 | 0 |
| | **Ours (3B)** | 4.112 | 54.175 | 0.288 | 7.992 | 38.094 | **1343** | **4.47 (1.4×)** | **27.7%** |
| | SeedVR2-3B | 4.687 | 51.09 | 0.295 | 8.176 | 38.1 | 2001 | 6.41 | 0 % |
| | SeedVR2-7B | 4.948 | 45.76 | 0.257 | 7.236 | 41.332 | 3139 | 7.74 | 0% |
| | **Ours (3B, one-step)** | **4.3449** | **57.783** | **0.324** | **8.199** | **39.220** | **1343** | **4.47 (1.4×)** | **27.7%** |
| AIGC30 | SeedVR-3B | **3.655** | 64.4 | **0.589** | 12.9 | **25.3** | 2001 | 6.41 | 0% |
| | SeedVR-7B | 4.053 | 64.26 | 0.564 | **16.47** | 35.7 | 3139 | 7.74 | 0% |
| | **Ours (3B)** | 3.99 | **65.14** | 0.575 | 11.9 | 35.89 | **1135** | **3.85 (1.6×)** | **39.6%** |
| | SeedVR2-3B | 3.801 | 62.99 | 0.561 | 15.77 | **28.021** | 2001 | 6.41 | 0% |
| | SeedVR2-7B | 4.138 | 65.09 | 0.574 | 13.4 | 39.14 | 3139 | 7.74 | 0% |
| | **Ours (3B, one-step)** | **3.6628** | **67.218** | **0.581** | **16.021** | 29.534 | **1135** | **3.85 (1.6×)** | **39.6%** |

layer on $P$, the output $P'_{\mathrm{unskip}}$ is composed with $P_{\mathrm{skip}}$ using $M$, resulting in a mixed sequence of processed and unchanged patches.

**Handling Window Attention.** Our DiT uses shifted window attention, which partitions the input into non-overlapping windows at each layer. In this case, the rotary positional encodings are not defined with respect to the entire input sequence length, but instead are based on the window size. To handle this with the SkipSR mask, we first assign each patch in $P$ to a window as normal, then apply the mask, resulting in $P_{\mathrm{unskip}}$. Each patch simply $P$ keeps its original window assignment from $P$, as shown in the inset of Figure 3. Although each window now has an imbalanced number of tokens, this is handled natively by FlashAttention and NaViT, while preserving rotary positional embeddings dependent on a fixed window size.

**Training.** To train the mask predictor, we freeze the VAE encoder and train on high-resolution images and videos. For the main super-resolution model, we follow standard practice and train on a mixture of images and videos (Chen et al., 2025; Wang et al., 2025b), using standard diffusion training.

Furthermore, while standard diffusion transformers are multi-step, several works accelerate these models by reducing the number of steps required to just one step. We train a one-step version of SkipSR to measure against these approaches, based on the procedure introduced in APT (Lin et al., 2025). We apply progressive distillation (Salimans & Ho, 2022) to distill our model to reduce one-step, then adversarially post-train it using a transformer-based discriminator to restore sharpness, resulting in a one-step model that matches the quality of the original.

## 5 EXPERIMENTS

**Implementation Details.** Models are initialized from public SeedVR/SeedVR2 (3B/7B) checkpoints and fine-tuned jointly with the same data and objectives as the baselines. Training uses 40 NVIDIA A100-80G GPUs with batches of about 100 frames at 720p, using sequence (Korthikanti et al., 2023) and data parallelism (Li et al., 2020). We use the same data for training as Wang et al. (2025b). The mask predictor is trained on the same data as the super-resolution model with 8 GPUs for 10k iterations, as the mask predictor converges significantly faster due to its smaller size.

**Experimental Settings.** Our goal is to simply compare the generation quality of the base model (SeedVR) on real-world videos while reducing wall-clock time. We conduct our main evaluation on the VideoLQ (Chan et al., 2022) and our AIGC30 datasets, reporting commonly used reference-free quality metrics such as NIQE (Mittal et al., 2013)c, CLIP-IQA (Wang et al., 2023a), MUSIQ (Ke et al., 2021) and DOVER (Wu et al., 2023). Efficiency is measured by runtime and skipped token fraction (token%). All comparisons use unmodified SeedVR / SeedVR2 with our masked

| Methods-{Steps} | Speed (seconds) | Overall Quality |
|---|---|---|
| SeedVR-3B-50 | 320 | +4% |
| DOVE-1 | 14.1 | -18% |
| SeedVR2-3B-1 | 6.41 | 0% |
| **Ours-3B-50** | **3.5** | **+7%** |

(a) **User Study.** Our method is preferred by users relative to SeedVR2 while also being significantly faster.

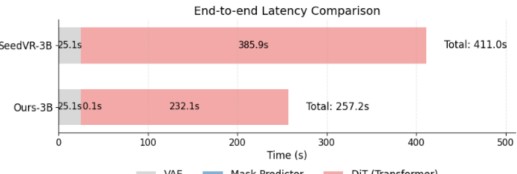

(b) **Latency.** The end-to-end generation latency for the transformer is significantly improved by SkipSR, reducing the total time by 60% while the mask predictor adds negligible overhead.

Table 3: **Results on Synthetic VSR Benchmarks.** SkipSR matches the performance of other methods on heavily degraded benchmarks. Best numbers are **blue**; second best are red.

| Datasets | Metrics | RealViformer | MGLD-VSR | STAR | DOVE | SeedVR-7B | SeedVR2-3B | SeedVR2-7B | SkipSR-3B *(Ours)* |
|---|---|---|---|---|---|---|---|---|---|
| SPMCS | PSNR ↑ | **24.185** | 23.41 | 22.58 | 23.11 | 20.78 | 22.97 | 22.90 | 23.9892 |
| | SSIM ↑ | 0.663 | 0.633 | 0.609 | 0.621 | 0.575 | 0.646 | 0.638 | **0.6807** |
| | LPIPS ↓ | 0.378 | 0.369 | 0.420 | 0.288 | 0.395 | 0.306 | 0.322 | **0.2865** |
| | DISTS ↓ | 0.186 | 0.166 | 0.229 | 0.171 | 0.166 | 0.131 | 0.134 | **0.1234** |
| UDM10 | PSNR ↑ | **26.70** | 26.11 | 24.66 | 26.48 | 24.29 | 25.61 | 26.26 | 26.5243 |
| | SSIM ↑ | 0.796 | 0.772 | 0.747 | 0.783 | 0.731 | 0.784 | **0.798** | 0.788 |
| | LPIPS ↓ | 0.285 | 0.273 | 0.359 | 0.270 | 0.264 | 0.218 | **0.203** | 0.2429 |
| | DISTS ↓ | 0.166 | 0.144 | 0.195 | 0.149 | 0.124 | 0.106 | **0.101** | 0.117 |
| REDS30 | PSNR ↑ | **23.34** | 22.74 | 22.04 | 22.11 | 21.74 | 21.90 | 22.27 | 22.4298 |
| | SSIM ↑ | **0.615** | 0.578 | 0.593 | 22.67 | 0.596 | 0.598 | 0.606 | 0.6093 |
| | LPIPS ↓ | 0.328 | 0.271 | 0.487 | 0.277 | 0.340 | 0.350 | 0.337 | **0.2682** |
| | DISTS ↓ | 0.154 | **0.097** | 0.229 | 0.106 | 0.122 | 0.135 | 0.127 | 0.1081 |
| YouHQ40 | PSNR ↑ | **23.26** | 22.62 | 22.15 | 24.3 | 20.60 | 22.10 | 22.46 | 23.1495 |
| | SSIM ↑ | 0.606 | 0.576 | 0.575 | 0.674 | 0.546 | 0.595 | 0.600 | **0.6277** |
| | LPIPS ↓ | 0.362 | 0.356 | 0.451 | 0.299 | 0.323 | 0.284 | 0.274 | **0.2361** |
| | DISTS ↓ | 0.193 | 0.166 | 0.213 | 0.148 | 0.134 | 0.122 | 0.110 | **0.1007** |

variants initialized from the same checkpoints and trained using the same procedure. Though not the main focus of our work, we also measure restoration ability on heavily degraded synthetic benchmarks, namely SPMCS (Yi et al., 2019), UDM10 (Yi et al., 2020), REDS30 (Nah et al., 2019), and YouHQ40, where we compute PSNR, SSIM, LPIPS and DISTS.

## 5.1 MAIN RESULTS

Since SkipSR is a general way to accelerate video SR, our goal is to match the performance of existing diffusion-based methods with minimal loss in quality while achieving a significant speed-up. We compare against the state-of-the-art diffusion methods, including other prior methods for completeness. Our main results on super-resolution benchmarks are in Table 2, where we evaluated SkipSR on real-world and AI-generated video SR test sets. In general, SkipSR matches or exceeds SeedVR's performance in video quality metrics, while significantly reducing the total time required, reducing it by 40% in VideoLQ and 60% on AIGC-30. We also match the performance of SeedVR2 in one-step generation, demonstrating the versatility of our framework and its potential applicability to cascaded diffusion. We also measure the end-to-end latency of the system in Figure 4b. The computation time is dominated by the diffusion transformer, with the VAE taking around 30s total and the mask predictor adding 60ms of overhead. The mask predictor also adds 23M parameters and 22MB of memory to the base model's 3B parameters and 13GB memory, incurring little cost overall. We find that on average, SkipSR reduces the overall end-to-end generation time by 60%, more than 2 minutes, strongly supporting its general applicability.

**User Study.** User studies are crucial for evaluating generative outputs. We conducted a GSB study with three domain experts, asking each to decide whether the samples are better, same, or worse. The preference score is calculated as $\frac{G-B}{G+S+B}$, which ranges from $-100\%$ to $100\%$. We randomly select 25 samples from VideoLQ and AIGC-30 and ask subjects to evaluate the overall quality of

Table 4: **Predicted Mask Analysis.** We repeat the analysis in Table 4 with the learned predictor and find that we generally maintain PSNR, but retain slightly more tokens.

| Dataset | Resolution | PSNR↑ | SSIM↑ | Skipped (%)↓ |
|---|---|---|---|---|
| VBench | 720p | 43.67 (+1.4) | 0.980 (−0.008) | 58.1 (-2.9) |
| VideoLQ | 720p | 43.87 (−2.7) | 0.991 (−0.001) | 27.7 (-3.5) |
| YouHQ-40 (clean) | 720p | 44.52 (-2.1) | 0.988 (-0.006) | 8.6 (-6.1) |
| AIGC-30 | 720p | 42.03 (−1.1) | 0.991 (-0.003) | 39.6 (-2.3) |
| VideoLQ | 1080p | 43.87 (−2.7) | 0.991 (−0.001) | 33.4 (-3.3) |
| AIGC-30 | 1080p | 43.24 (−1.2) | 0.993 (+0.002) | 42.9 (-2.0) |

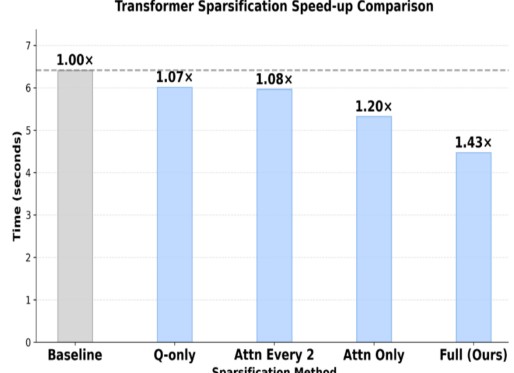

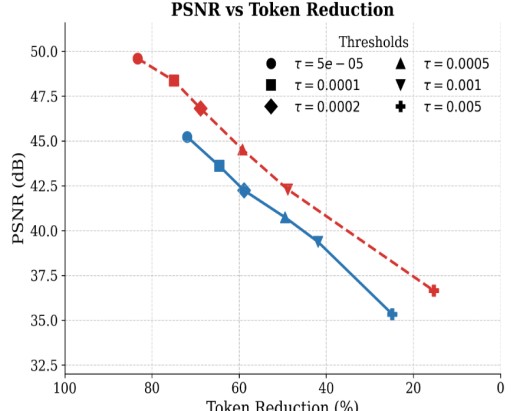

(a) **Sparsity mechanism.** We measure the speed of routing around the model entirely compared to other design choices, such as only masking the attention, using interspersed global layers, or only masking the attention queries. Ours is significantly faster, while maintainng quality.

(b) **Threshold effect.** As we increase the threshold $\tau$, the reduction increases at the cost of maximum PSNR. $\tau = 0.0002$ is a reasonable choice for token reduction while maintaining perceptual quality.

our generations compared to the baseline, SeedVR2. As shown in Table 4a, we match SeedVR-3B and SeedVR2, and significantly outperform DOVE, supporting our claim that we can accelerate the model without losing visual quality as perceived by users.

**Synthetic Benchmarks.** Synthetic video restoration benchmarks are created by applying small degradations to the entire frame, making them unsuitable for our core hypothesis, that upsampled regions from low-resolution videos map well onto their high-resolution outputs. Achieving state-of-the-art performance on these is not the main focus of this work; we are emphatically focused on general acceleration without significant loss of quality. However, we include the results of these benchmarks in Table 3 for completeness. Our model performs surprisingly well, matching the performance of comparable models, such as SeedVR, DOVE, and SeedVR2.

## 5.2 ANALYSIS

**Mask Predictor.** We repeat the analysis of the oracle experiment in Section 3 using our learned mask predictor, comparing the theoretical maximum performance using our predicted mask in absence of the ground-truth high-resolution video with that of the oracle using the ground-truth. The results in Table 4 demonstrate that our mask performs well: it does not quite reach the theoretical optimum but still skips a significant proportion of patches while maintaining visual fidelity. Notably, the difference between swapping with our predicted mask and the ground-truth high-resolution is still consistently above 40 PSNR, which is an essentially imperceptible difference.

**Profiling and Speed Design Choices.** We measure the efficiency of other potential sparsity mechanisms in Figure 5a. Most works address the attention layer; we demonstrate that this loses 23%

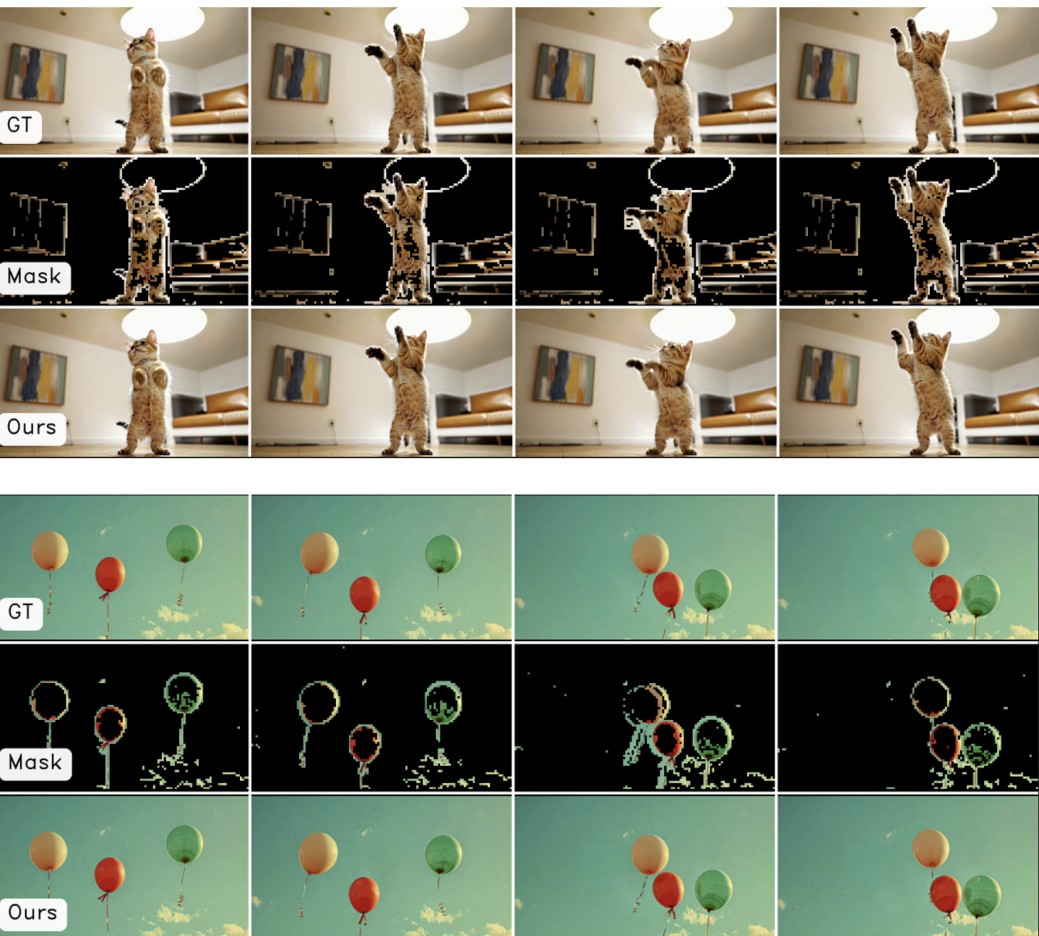

Figure 6: **Visual Comparison.** In the above two examples, the ground truth is on top, followed by the predicted mask, and our output. We produce perceptually indistinguishable results while only refining a small subset of the input video. Additional results are on our project page.

speed. Similarly, strategies like only masking out the attention queries or interspersing sparse and dense layers are not much faster than the baseline. On the other hand, our method achieves the largest speed-up while maintaining quality.

**Threshold Effect.** Finally, we analyze the effect of varying the threshold $\tau$ that defines the boundary between simple and complex patches in Figure 5b. We measure the PSNR compared to the original image for both AIGC and VideoLQ using the oracle mask, as well as the fraction of tokens removed by this procedure. Since the mask is dependent on the input data, the curves are offset from each other, but follow the same general trend: as we increase the threshold, the PSNR consistently decreases as more and more complex patches are skipped rather than refined. From the plot, we can see that $\tau = 0.0002$ represents a reasonable balance between fidelity and token reduction, as it is paramount that any acceleration does not come at a cost to visual quality.

## 6 CONCLUSION

This paper tackles the acceleration of video super-resolution by focusing refinement only on complex regions that actually require it, rather than uniformly upscaling the entire input. We find that such regions make up a substantial fraction of video, and that using a cheap upsampling operation on them rather than expensive super-resolution can maintain quality. We propose SkipSR, which identifies these regions from the low-resolution input and routes them entirely around the transformer,

composing them with the upscaled output afterward. Implementing our simple strategy enables significant acceleration, reducing the diffusion time by $1.8\times$ without visible loss in quality and the end-to-end generation time by $1.6\times$.

**Limitations.** A limitation of SkipSR is its lack of acceleration on video restoration tasks with severe degradations, where we are unable to skip patches due to widespread corruptions. In general, SkipSR does not provide significant speed-ups on videos with extremely crowded scenes or camera jitter, but we believe its strong performance on more typical videos is well worth the tradeoff.

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

## A    IMPLEMENTATION DETAILS

In this section, we provide more details on the preliminiaries and implementation details for SkipSR. These are based on prior works, **note about referring to other works**

**Diffusion-Based Super-Resolution.**

**Causal VAE.**   As described in the main text, SkipSR follows the standard latent-diffusion paradigm, where we encode pixel inputs using a variational autoencoder (VAE) into a compressed latent space, perform diffusion, then decode back into pixel space. Prior works **CITE** of ten use an inflated image-based VAE: **explain**.

However, we employ the causal VAE introduced in SeedVR. The key difference is that rather than ...

## B    ADDITIONAL EXPERIMENTS

**Mask Predictor Analysis**   We include additional analysis on the performance of our learned mask predictor in Table **add Ref**. We measure the precision, recall, mAP, and F1 score across different datasets, and visualize the performance of the predictor **add some more**

**Image Super-Resolution Results**   The main focus of SkipSR is optimizing video super-resolution, since videos are much larger than images and thus are more computationally challenging. However, since SkipSR is trained on a mixture of images and videos, it is capable of performing image super-resolution as well. We include results of SkipSR on DIV2K in Table , comparing with SeedVR, SeedVR2, and other notable upscaling methods. Similar to the video domain, SkipSR matches the performance of current methods while skipping a significant fraction of the tokens, **add mo analysis**

## C    ADDITIONAL VISUALIZATIONS

We encourage readers interested in video examples of our work to visit our our [project page](https://clamsoup97.github.io/anonymous-projects/skipsr/), where several demos are shown. We also include some more examples here. Our visualizations show that the mask predictor consistently identifies simple patches, and that SkipSR produces excellent super-resolution quality despite skipping on average 40% of the input tokens.

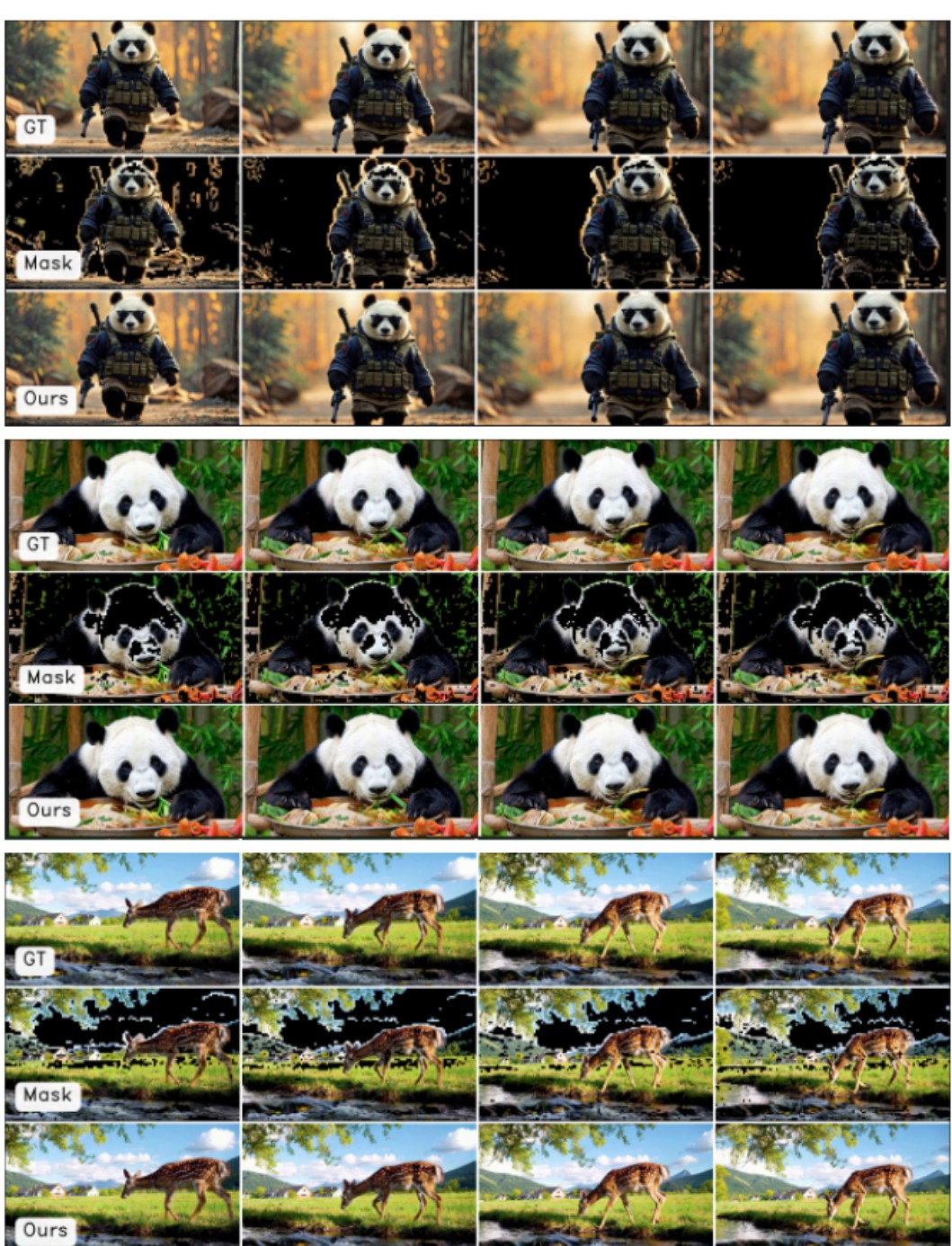

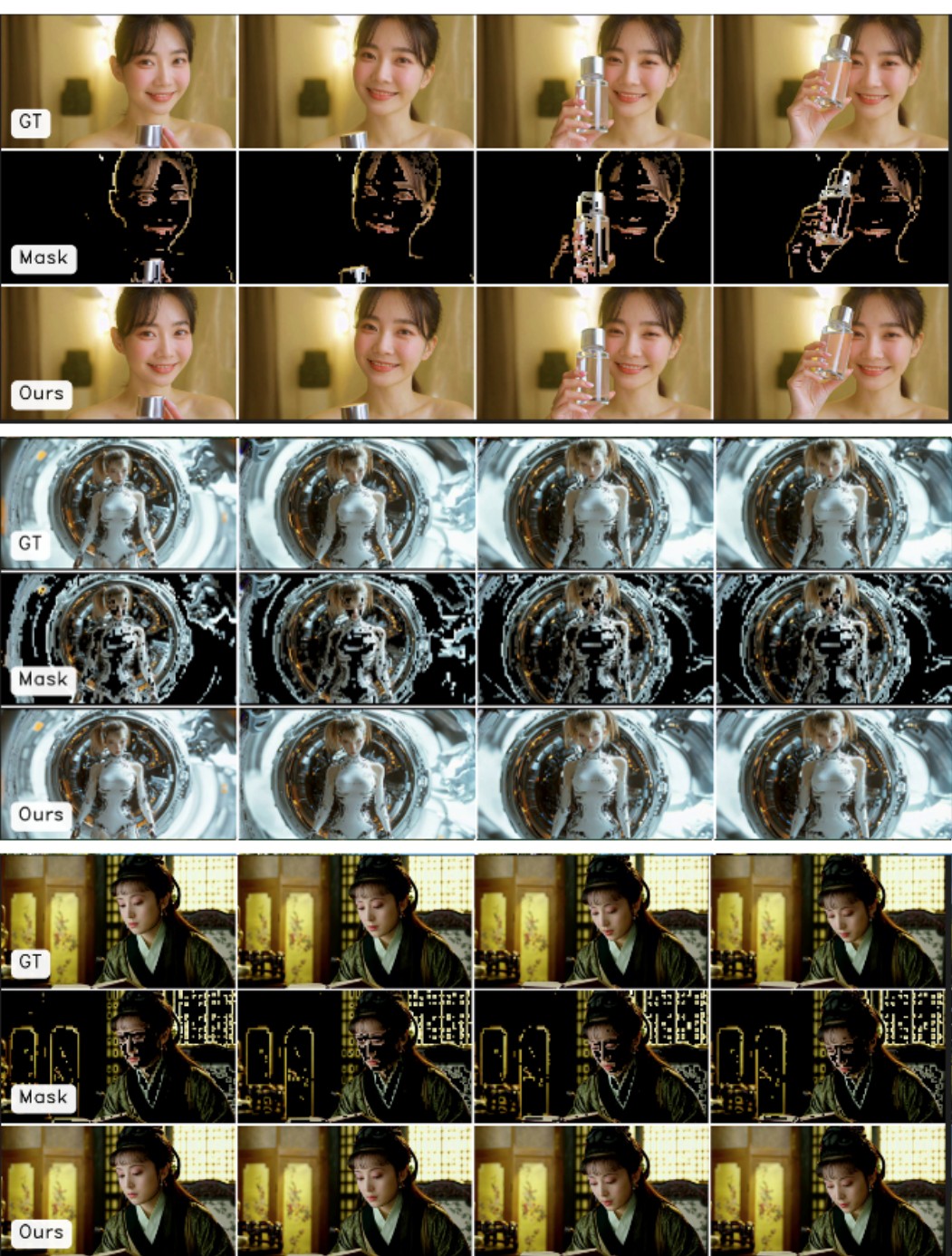

