# OpenReview forum: "SkipSR: Faster Super-Resolution with Token Skipping"
_ICLR.cc/2026/Conference — ICLR 2026 Conference Desk Rejected Submission_

### Official Review · Reviewer_BvQa · 2025-10-20

**Soundness:** 3
**Presentation:** 2
**Contribution:** 2
**Rating:** 6
**Confidence:** 5

**Summary:**

This paper proposes SkipSR to focus on refining complex regions only and skip simple regions by directly upsampling in pixel space. To determine the simple regions, it determines a threshold offline and  trains a light-weight predictor module. Detailed experiments in multi-step and one-step setting demonstrate the efficiency of the proposed method in cascaded video super-resolution scenario.

**Strengths:**

* The authors observed an unnoticed problem: smooth regions and complex regions were allocated the same amount of computational power. They proposed a simple and effective method to address it.
* To implement their method to open-source framework, they introduced several modifications to adapt to a swin-based DiT backbone.
* Detailed experiments, including qualitative, quantitative, user study and video results, demonstrate its effectiveness.

**Weaknesses:**

* ● My primary concern lies in the simple supervision of this predictor. In real-world VSR, the input videos often contain high-frequency degradation patterns, which conflicts the assumption in this paper. Also, no real-world results are shown in the video demo. The authors should explain why the current objective remains effective in real-world scenarios. Otherwise, the authors should consider to restrict the scope of SkipSR to cascaded video super-resolution (for AIGC videos only).
* SkipSR is implemented on a window-based DiT backbone and the results are mixed with upsampled input by complexity mask. Thus, I am concerned about the seam problem and color difference between adjacent areas in the generated results. The authors should indicate the potential risks associated with this method or explain why such issues do not exist.
* The motivation of skip-aware RoPE is unclear. By using original RoPE setting, each unskipped token can access other tokens in the 3D attention module. By adapting with the flash attention, the proposed method can seamlessly integrate into its DiT backbone. Thus, modifying the RoPE is confusing and unnecessary.

**Questions:**

* More details about the mask predictor should be added, including the selection of the training videos, the convergence speed and its robustness against diverse testing scenarios.
* A formula is missed in line 262, and the part of skip-aware RoPE is unclear now.
* The authors are recommended to provide more uncompressed video results, including real-world and aigc scenarios.

---

> ### Author Response · Authors · 2025-11-25
> **Response: Reviewer BvQa**
>
> Thank you for your thorough and helpful review of our paper! We appreciate that you found our paper to address an unnoticed problem, and that our solution was effective. We address your concerns below:
>
> __Weakness: Simple Supervision for Predictor and Applicability to Real-World Videos.__
>
> Thanks for pointing this out. Our mask predictor is trained on ground-truth obtained by downsampling, then upsampling input videos and images, then computing their mean-squared error to identify regions that can be masked.
> As the reviewer noted, our method is indeed best suited for cascaded diffusion pipelines, but we find it works well on challenging real-world scenarios also.
> To show this, we have added some visual samples from VideoLQ, a challenging real-world input dataset, to our anonymized project page. We also included an evaluation of VideoLQ in Table 2, where SkipSR achieves strong restorative ability with large speedups. Both the visual examples and the quantitative results demonstrate that SkipSR works well, even on real-world results. However, as we note, this does not extend to synthetically corrupted benchmarks such as YouHQ-40. On these synthetic benchmarks, we match the restorative performance of prior work, but are no faster. We believe this is an acceptable tradeoff, since the type of degradation occurring in VideoLQ is much more commonplace than in synthetic benchmarks.
>
> __Weakness: Possibility of seams from windowing and masking.__
>
> There is certainly a possibility that seam issues can occur. We identified a failure case where seams are visible on the VideoLQ dataset, which we have added to the “Failure Cases” section of the updated project page. However, in general, this does not cause visible issues, as evidenced by our User Study on page 7 and the qualitative results on our project page. We believe there are three main reasons that seam issues are generally mitigated. First, the mask predictor is explicitly trained based on ground truth masks that minimize the PSNR between ground-truth videos and the versions with mask-based upsampling. As shown in Table 1, if the mask prediction is good enough, the difference should be imperceptible (50+ PSNR), and there should not be regions where a seam would occur. Next, the conditioning information in diffusion-based super resolution is extremely helpful: for regions on the edge of a masked-out area, the input condition usually provides enough information for it to get the intensity correct. Finally, our base super-resolution models are already strong and based on large pre-trained checkpoints such as SeedVR and SeedVR2, which use adaptive window attention to mitigate seams. We believe the combination of these three factors enables the model to generally avoid such issues.
>
> __Weakness: Skip-Aware Rope motivation is unclear.__
>
> We apologize for the confusion. It is true that basic RoPE can be applied seamlessly with FlashAttention to our method - after all, we are simply passing in a set of variable-length sequences of tokens. However, this does not work for a crucial reason.RoPE tells the model where each patch is, and consequently, how far apart they are in space and time. The relative positions of tokens within a window crucially do not change after masking: if we mask out alternate tokens, the resulting tokens are now 2 tokens apart from each other. As a result, the rotations that are applied to each token in the sequence need to be those that correspond to their original position in the full sequence, not their new position in the flattened, masked sequence. This is why we need skip-aware RoPE; without this, the model would not have information about where tokens are relative to each other - essentially, it would not know about the new “holes” in the token arrangement after masking. As we demonstrate in the inset of Figure 3, after masking, the tokens can be separated in space and time, and the positional encodings need to take this into account. We have clarified our explanation on Page 5, Lines 259-269 of the updated text.
>
> __Question: More details about the mask predictor should be added.__
>
> Thanks for this! We have updated the text to include more details about the training and its robustness on Page 6, Lines 316-317. In particular, we train the mask predictor on the same dataset that we use for training the full end-to-end super-resolution model. The model converges relatively quickly, in about 10000 iterations, and we tested it on a wide variety of scenarios. We have also added more visualizations to the project page to help reviewers better understand the mask predictor’s performance on more dynamic and real-world videos.
>
> __Question: Missing Formula on Line 262.__
>
> Thank you for catching this! We apologize, and we have added in the formula and updated the relevant text to make the explanation clearer.
>
> If we have sufficiently addressed your concerns, would you consider raising your score? We are glad to answer any further questions as well.

---

### Official Review · Reviewer_Hc6Y · 2025-10-22

**Soundness:** 3
**Presentation:** 3
**Contribution:** 3
**Rating:** 8
**Confidence:** 4

**Summary:**

SkipSR is a framework that accelerates Video Super-Resolution (VSR) by skipping visually simple tokens for the diffusion loop and just using their interpolation (simple tokens are defined by undercutting a threshold for the reconstruction error of the VAE and interpolation) instead. By doing so, they achieve substantial speedups (if the videos are overall not heavily degraded), which is an interesting idea and important research direction for the community.

**Strengths:**

- Although the selection of visually important areas to apply SR models to is not entirely novel, this paper goes a step further than previous work by completely bypassing transformer computation for certain regions to achieve substantial runtime improvement.
- The technical quality of the work is strong and thorough. The authors motivate their design through empirical analysis, showing that large portions of typical videos consist of low-frequency, simple regions that can be reconstructed with minimal loss using cheap upsampling.
- The paper is clearly structured and easy to follow.

**Weaknesses:**

- The authors highlighted that themselves: SkipSR time improvement is limited to clean inputs with skippable patches.
- Dependence on a heuristic threshold and empirical tuning. The threshold sounds a bit arbitrary and should be analyzed further. Also, an analysis on how good the mask predictor is, when does it fail and so on, would be appreciated.

**Questions:**

- The paper uses a relatively lightweight 3D convolutional predictor to identify skippable patches. Could the authors clarify how sensitive SkipSR’s performance is to the predictor design?
- Have you experimented with 2D predictors (spatial only) or different temporal receptive fields to assess whether temporal context meaningfully improves mask accuracy?
- How stable are the predicted masks across frames in dynamic scenes? If the mask varies rapidly between adjacent frames, could this introduce subtle flicker or temporal inconsistencies in the reconstructed video?
- The skip threshold τ = 0.0002 appears empirically tuned. Do the authors foresee a way to make this parameter adaptive (e.g., through training the mask predictor on the fly and using the mask sparsity as additional regularization)?

---

> ### Author Response · Authors · 2025-11-25
> **Response: Reviewer Hc6Y**
>
> Thank you for your positive review! We appreciate that you found SkipSR to be technically strong and thorough. We address some of the concerns you highlighted below.
>
> __Weakness: Limited to clean inputs.__
> We agree with the reviewer’s assessment - SkipSR only improves if there are indeed clean, skippable patches. We added more results on real-world video super-resolution on our project page, and show that our method works on more dynamic videos with high-speed motion and camera shake as well. However, we agree that for extremely corrupted videos, SkipSR will provide no speedup. As the reviewer notes, we think this is reasonable given that a nontrivial fraction of real-world and AI generated video are simple and contain skippable patches, as measured from our empirical experiments, and SkipSR can be especially effective for cascaded diffusion pipelines.
>
> __Weakness: Threshold Dependence and Mask Predictor Failure Analysis.__
>
> This is a reasonable observation. The threshold is used for generating ground-truth supervision for the predictor, and we systematically measure its effect on PSNR  in Figure 5b on page 8.  While it is true that the threshold needs to be empirically tuned, it was chosen to balance the tradeoff between reducing tokens (increasing speed) and PSNR (maintaining visual fidelity).
>
> In general, the mask predictor is trained to be conservative: rather than masking too many tokens, it errs on the conservative side. We have added more visual examples to the project page, demonstrating its performance on complex and dynamic scenes, as well as two failure cases. We observe that the predictor tends to flicker on the same region over time, even when the change is imperceptible, suggesting that the predictor’s temporal consistency could be improved. We believe that future work can use more bespoke architectures and losses to improve this part of the model.
>
> __Question: 2D Spatial Predictor.__
>
> This is an excellent question. SkipSR's overall performance is dependent on the predictor performance, and varying the design does indeed change the result. Simpler architectures, such as a one or two-layer MLP, work reasonably well but do not perform as well as CNNs. We also tried a 2D CNN, as the reviewer suggests, but found that a 3D CNN is a simple change that reduces flickering significantly more than 2D.
>
> __Question: Can we learn an adaptive mask predictor without a threshold, e.g using a sparsity loss?__
>
> In early experiments, we did try an idea similar to this. However, training a mask predictor without specifying a target sparsity fraction and using a sparsity regularizer tends to either converge to a trivial solution (mask retains everything) or collapse to zero (mask removes everything) when training jointly with the super-resolution task. This observation has been made in several prior works concerning mixture-of-experts and using load-balancing loss[1], or learned language tokenization methods [2]. This would be an excellent avenue for future work, since a more data-driven solution would likely be helpful across a range of tasks in the community beyond efficient super-resolution.
>
> Thank you again, and please feel free to respond with more questions or comments - we are happy to discuss our work further and alleviate any concerns you may have!
>
>
>
> [1] Shazeer, Noam, Azalia Mirhoseini, Krzysztof Maziarz, Andy Davis, Quoc Le, Geoffrey Hinton, and Jeff Dean. "Outrageously large neural networks: The sparsely-gated mixture-of-experts layer." arXiv preprint arXiv:1701.06538 (2017).
>
> [2] Hwang, Sukjun, Brandon Wang, and Albert Gu. "Dynamic chunking for end-to-end hierarchical sequence modeling." arXiv preprint arXiv:2507.07955 (2025).

---

> > ### Comment · Reviewer_Hc6Y · 2025-11-27
> >
> > I thank the authors for the clarifying answers. I will keep my score as it is already opting for acceptance.

---

### Official Review · Reviewer_YAoC · 2025-10-31

**Soundness:** 2
**Presentation:** 3
**Contribution:** 2
**Rating:** 4
**Confidence:** 5

**Summary:**

This paper introduces SkipSR – a method to accelerate video SR by skipping transformer processing in “simple” regions like sky or simple background. These skipped regions are upsampled via bilinear interpolation. The authors claim the method achieves up to 60% faster end-to-end latency on 720p videos with no perceptible quality loss.

**Strengths:**

1. The idea is original in its application to modern video diffusion transformers.
2. Experiments are broad, covering multiple datasets, resolutions, and model types.
3. The speedups are substantial and significant for practical contribution for real-world applications.

**Weaknesses:**

1. The analysis in Table 1 shows that on heavily corrupted videos, the skippable patch percentage drops to 0.9%, resulting in no speedup. This suggests the method's utility is highly dataset-dependent and may not generalize well to challenging restoration tasks.
2. The mask predictor is crucial, yet its design is too simple. There is no ablation study on its architecture, nor a comparison to potentially more sophisticated segmentation networks
3. The practical overhead of the masking operation itself is not profiled in detail. The claim on Page 7 that the mask predictor adds "negligible overhead" is stated but not quantitatively broken down, which is important for a method whose value is entirely in net speedup.
4. The paper does not deeply explore the trade-off between mask accuracy and computational overhead in more dynamic or complex scenes.

**Questions:**

1. Could the masking strategy be adapted dynamically per frame or scene to improve robustness?
2. Have you considered combining SkipSR with other efficiency methods like distillation or quantization for further speedup?
3. Given the significant performance drop on corrupted videos (YouHQ-40 corrupted), what is a realistic estimate of the fraction of real-world video content where SkipSR would provide a visible speedup?

---

> ### Author Response · Authors · 2025-11-25
> **Response: Reviewer YAoC (1/2)**
>
> Thank you for your insightful and helpful review. We are glad that you found our work to be original and that we deliver substantial speedups. We address your concerns below:
>
> __Weakness: not applicable to synthetically corrupted video.__
>
> The reviewer is correct: the speed-up from SkipSR is highly dataset dependent. It provides a strong speedup on relatively simple videos, which contain large regions that do not change too much over time. This assumption generally holds for AI-generated videos and for real-world videos, as evidenced by Table 1. We have updated our project page to include real-world video examples with significantly more dynamic scenes and camera shaking, demonstrating our method's applicability to complex videos.
>
> As we noted on Lines 178-180, this assumption does not hold on datasets where the entire video is subject to synthetic high-frequency degradations, such as YouHQ-40 and other artificial restoration tasks. However, SkipSR maintains strong restorative performance on these videos, as shown in Table 3; it is just not faster than prior works. As a result, SkipSR is at least as fast as any other diffusion-based super-resolution method, while significantly faster on simpler videos. We believe the core assumption of our paper is reasonable: as Reviewers 1 and 3 note, SkipSR can be applied to cascaded diffusion pipelines, to stand-alone upscalers, and to real-world videos that are much more common than synthetic benchmarks.
>
> __Weakness: mask predictor simplicity.__
>
> The mask predictor is certainly crucial, but we disagree that its design is “too simple”.  A strong mask predictor for SkipSR should be a high-quality binary classifier, *lightweight*, and *efficient*. To draw an analogy, token routing mechanisms in mixture-of-experts models are typically just a single MLP layer! In SkipSR, we certainly could use a large transformer-based segmentation network, but this would add significant overhead, negating any efficiency gains. While a larger and more bespoke network would likely improve the classification performance of the mask predictor slightly, we think that this would detract from the focus of the paper and that this can be improved further in future work.
>
> __Weakness: Masking operation overhead.__
>
> We apologize for the lack of detail, and have updated the text to include the precise memory, parameter count and timing of the mask predictor on Lines 365-367 of Page 7. Please let us know if you would like additional detail: we are happy to provide it.
>
> __Weakness: Tradeoff between mask accuracy and computational overhead in dynamic scenes.__
>
> We are not certain we understand this point. In SkipSR, the computational overhead is totally agnostic to the scene content - predicting the mask takes the same time no matter what the scene is. For more dynamic scenes where less content can be masked out, there will be no more overhead than in other scenes. We have updated the project page to include some visualizations of the mask predictor operating in more real-world and dynamic scenes. Would it be possible to clarify this point so that we can better understand and address your concern?
>
> __Question: Could the mask be dynamically adapted per frame or scene for robustness?__
>
> We are also uncertain we understand this question. SkipSR’s predicted mask is already dynamic - the fraction of skippable content changes based on the input video, and since we are using this for super-resolution, the entire video or image is available as input. As is visible on the demo project page, the mask changes constantly with frames and scenes in each video.  Would it be possible to clarify if we misunderstood the question? We would be glad to address your concerns if we did not.

---

> > ### Author Response · Authors · 2025-11-25
> > **Response: Reviewer YAoC (2/2)**
> >
> > __Question: Have you considered combining SkipSR with distillation or quantization for speedup?__
> >
> > These are excellent suggestions, and we believe that these would be valuable additions to SkipSR if deployed to a production pipeline. To train the one-step diffusion model, we did use progressive distillation, which delivers an additional speedup on top of the acceleration provided by SkipSR. However, we think that our paper’s scope should be limited to the discussing the notion of token skipping, and how it impacts video and image super-resolution rather than introducing and focusing on other techniques for speedup.
> >
> > __Question: Given the significant performance drop on corrupted videos, what is a realistic estimate of fraction of skippable real-world content?__
> >
> > To clarify, SkipSR still maintains the high restorative ability of prior methods on synthetically corrupted videos; it just does not provide a speed-up. From the results in Table 1, we can skip 30% of tokens on VideoLQ, a particularly challenging real-world dynamic-scene video benchmark, and 40+% on AIGC benchmarks such as VBench and AIGC-28. A conservative estimate might be 30%, which still leads to a 50% speed-up - this would still be very impactful and significant for diffusion-based super-resolution.
> >
> > Thank you again for your helpful points and feedback. If we have addressed your concerns, would you consider raising your score for the paper?

---

### Note · Program_Chairs · 2026-01-17
**Submission Desk Rejected by Program Chairs**

The following references in this submission do not refer to real documents and/or have major errors in bibliographic information:

 Jiahui Xu, Jingyun Liang, Kai Zhang, and Luc Van Gool. Star: Structure-aware diffusion for video super-resolution. arXiv preprint arXiv:2310.07894, 2023.